# Relaxed Scheduling for Scalable Belief Propagation

**Vitaly Aksenov**
ITMO University
aksenov.vitaly@gmail.com

**Dan Alistarh**
IST Austria
dan.alistarh@ist.ac.at

**Janne H. Korhonen**
IST Austria
janne.korhonen@ist.ac.at

## Abstract

The ability to leverage large-scale hardware parallelism has been one of the key enablers of the accelerated recent progress in machine learning. Consequently, there has been considerable effort invested into developing efficient parallel variants of classic machine learning algorithms. However, despite the wealth of knowledge on parallelization, some classic machine learning algorithms often prove hard to parallelize efficiently while maintaining convergence.

In this paper, we focus on efficient parallel algorithms for the key machine learning task of inference on graphical models, in particular on the fundamental belief propagation algorithm. We address the challenge of efficiently parallelizing this classic paradigm by showing how to leverage scalable relaxed schedulers in this context. We present an extensive empirical study, showing that our approach outperforms previous parallel belief propagation implementations both in terms of scalability and in terms of wall-clock convergence time, on a range of practical applications.

## 1 Introduction

*Hardware parallelism* has been a key computational enabler for recent advances in machine learning, as it provides a way to reduce the processing time for the ever-increasing quantities of data required for training accurate models. Consequently, there has been considerable effort invested into developing efficient parallel variants of classic machine learning algorithms, e.g. [28, 22, 23, 24, 15].

In this paper, we will focus on efficient parallel algorithms for the fundamental task of *inference on graphical models*. The inference task in graphical models takes the form of *marginalisation*: we are given observations for a subset of the random variables, and the task is to compute the conditional distribution of one or a few variables of interest. The marginalization problem is known to be computationally intractable in general [10, 32, 9], but inexact heuristics are well-studied for practical inference tasks.

One popular heuristic for inference on graphical models is *belief propagation* [27], inspired by the exact dynamic programming algorithm for marginalization on trees. While belief propagation has no general approximation or even convergence guarantees, it has proven empirically successful in inference tasks, in particular in the context of decoding low-density parity check codes [8]. However, it remains poorly understood how to properly parallelize belief propagation.

**Parallelizing belief propagation.** To illustrate the challenges of parallelizing belief propagation, we will next give a simplified overview of the belief propagation algorithm, and refer the reader to Section 2 for full details. Belief propagation can be seen as a *message passing* or a *weight*

*update* algorithm. In brief, belief propagation operates over the underlying graph $G = (V, E)$ of the graphical model, maintaining a vector of real numbers called a *message* $\mu_{i \to j}$ for each ordered pair $(i, j)$ corresponding to an edge $\{i, j\} \in E$ (Fig. 1). The core of the algorithm is the *message update rule* which specifies how to update an outgoing message $\mu_{i \to j}$ at node $i$ based on the *other* incoming messages at node $i$; for the purposes of the present discussion, it is sufficient to view this as black box function $f$ over these other messages, leading to the update rule

$$\mu_{i \to j} \leftarrow f\big(\{\mu_{k \to i} \colon k \in N(i) \setminus \{j\}\}\big). \tag{1}$$

This update rule is applied to messages until the values of messages have converged to a stable solution, at which point the algorithm is said to have terminated.

Importantly, the message update rule does not specify *in which order* messages should be updated. The standard solution, called *synchronous belief propagation*, is to update all the message simultaneously. That is, in each global round $t = 1, 2, 3, \ldots$, given message values $\mu_{i \to j}^t$ for all pairs $(i, j)$, the new values $\mu_{i \to j}^{t+1}$ are computed as

$$\mu_{i \to j}^{t+1} \leftarrow f\big(\{\mu_{k \to i}^t \colon k \in N(i) \setminus \{j\}\}\big)$$

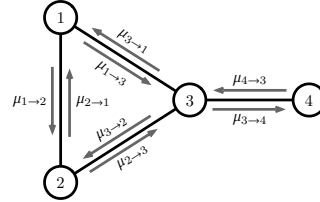

Figure 1: State of the belief propagation algorithm consist of two directed messages for each edge.

However, there is evidence that updating messages *one at a time* leads to faster and more reliable convergence [14]; in particular, various proposed *priority-based schedules*— schedules that try to prioritize message updates that would make 'more progress'—have proven empirically to converge with much fewer message updates than the synchronous schedule [14, 20, 37].

Having to execute updates in a strict priority order poses a challenge for efficient *parallel* implementations of belief propagation: while the synchronous schedule is naturally parallelizable, as all message updates can be done independently, the more efficient priority-based schedules are inherently sequential and thus seem difficult to parallelize. Accordingly, existing work on efficient parallel belief propagation has focused on designing custom schedules that try to import some features from the priority-based schedules while maintaining a degree of parallelism [15, 11].

**Our contributions.**  In this work, we address this challenge by studying how to efficiently parallelize any priority-based schedule for belief propagation. The key idea is that we can *relax* the priority-based schedules by allowing limited out-of-order execution, concretely implemented using a *relaxed scheduler*, as we will explain next.

Consider a belief propagation algorithm that schedules the message updates according to a priority function $r$ by always updating the message $\mu_{i \to j}$ with the highest priority $r(\mu_{i \to j})$ next; this framework captures existing priority-based schedules such as residual belief propagation [14] and its variants [20, 37]. Concretely, an iterative centralized version of this algorithm can be implemented by storing the messages in a priority queue $Q$, and iterating the following procedure:

    (1) Pop the top element for $Q$ to obtain the message $\mu_{i \to j}$ with highest priority $r(\mu_{i \to j})$.

    (2) Update message $\mu_{i \to j}$ following (1).

    (3) Update the priorities in $Q$ for messages affected by the update.

This template does not easily lend itself to efficient parallelization, as the priority queue $Q$ becomes a contention bottleneck. Previous work, e.g. [15, 11] investigated various heuristics for the parallel scheduling of updates in belief propagation, trading off increased parallelism with additional work in processing messages or even potential loss of convergence.

In this paper, we investigate an alternative approach, replacing the priority queue $Q$ with a *relaxed priority queue (scheduler)* to obtain a efficient parallel version of the above template. A *relaxed scheduler* [3, 1, 2, 5] is similar to a priority queue, but instead of guaranteeing that the *top* element is always returned first, it only guarantees to return *one of the top q elements by priority*, where $q$ is a variable parameter. Relaxed schedulers are popular in the context of parallel graph processing, e.g. [16, 26], and can induce non-trivial trade-offs between the degree of relaxation and the scalability of the underlying implementation, e.g. [1, 5].

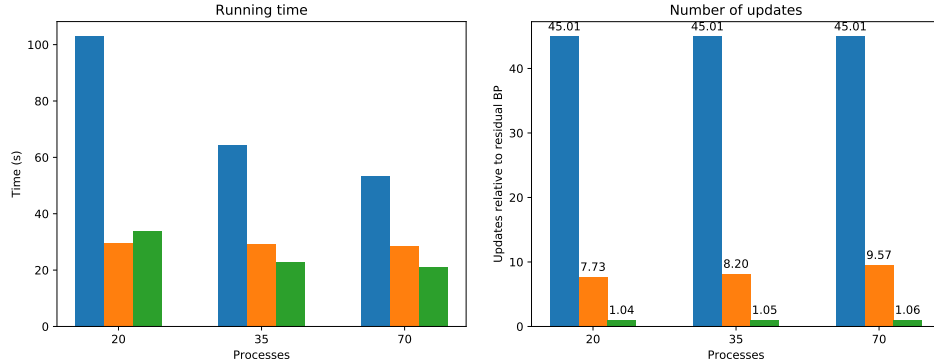

Figure 2: Running time (**left**) and number of updates until convergence (**right**) on a $1000 \times 1000$ Ising grid model on $p \in \{20, 35, 70\}$ processes. Included algorithms are synchronous belief propagation, residual splash belief propagation of Gonzalez et al. [15] with splash size 10, and our relaxed residual belief propagation. The number of updates is relative to sequential residual belief propagation.

For belief propagation, relaxed schedulers induce a *relaxed priority-based scheduling* of the messages, roughly following the original schedule but allowing for message updates to be performed out of order, with guarantees on the maximum degree of priority inversion. We investigate the scalability-convergence trade-off between the *degree of relaxation* in the scheduler, and the *convergence behavior* of the underlying algorithm, both theoretically and practically. In particular:

– We present a general scheme for parallelizing belief propagation schedules using relaxed schedulers with theoretical guarantees. While relaxed schedulers have been applied in other settings, and relaxed scheduling has been touched upon in belief propagation scheduling [15], no systematic study on relaxed belief propagation scheduling has been performed in prior work.
– We provide a theoretical analysis on the effects of relaxed scheduling for belief propagation on trees. We exhibit both positive results–instance classes where relaxation overhead is negligible–and negative results, i.e., worst-case instances where relaxation causes significant wasted work.

**Experimental evaluation.**   We implement our relaxed priority-based scheduling framework with a *Multiqueue* data structure [30] and instantiate it with several known priority-based schedules. In the benchmarks, we show that this framework gives state-of-the-art parallel scalability on a wide variety of Markov random field models. As expected, the relaxed priority-based schedules require slightly more message updates than their exact counterparts, but this performance overhead is offset by their better scalability. In particular, we highlight the fact that the relaxed version of the popular residual belief propagation algorithm has state-of-the art parallel scaling while requiring no tuning parameters, making it an attractive practical solution for belief propagation. This finding is illustrated in Figure 2, and further substantiated in Section 5.

**Full version.** Full version of this paper is available at https://arxiv.org/abs/2002.11505.

## 2   Preliminaries and Related Work

### 2.1   Belief Propagation

We consider marginalization in *pairwise Markov random fields*; one can equivalently consider factor graphs or Bayesian networks [39]. A pairwise Markov random field is defined by a set of random variables $X_1, X_2, \ldots, X_n$, a graph $G = (V, E)$ with $V = \{1, 2, \ldots, n\}$, and a set of *factors*

$$\psi_i \colon D_i \to \mathbb{R}^+ \qquad\qquad \text{for } i \in V,$$
$$\psi_{ij} \colon D_i \times D_j \to \mathbb{R}^+ \qquad\qquad \text{for } \{i, j\} \in E,$$

where $D_i$ denotes the domain of random variable $X_i$. The edge factors $\psi_{ij}$ represent the dependencies between the random variables, and the node factors $\psi_i$ represent a priori information about the individual random variables; the Markov random field defines a joint probability distribution on

$X = (X_1, X_2, \ldots, X_n)$ as

$$\Pr[X = x] \propto \prod_i \psi_i(x_i) \prod_{ij} \psi_{ij}(x_i, x_j),$$

where the 'proportional to' notation $\propto$ hides the normalization constant applied to the right-hand side to obtain a probability distribution. The marginalization problem is to compute the probabilities $\Pr[X_i = x]$ for a specified subset of variables; for convenience, we assume that any observations regarding the values of other variables are encoded in the node factor functions $\psi_i$.

Belief propagation is a message-passing algorithm; for each ordered pair $(i, j)$ such that $\{i, j\} \in E$, we maintain a *message* $\mu_{i \to j} \colon D_j \to \mathbb{R}$, and the algorithm iteratively updates these messages until the values (approximately) converge to a fixed point. On Markov random fields, the message update rule gives the new value of message $\mu_{i \to j}$ as a function of the old messages directed to node $i$ by

$$\mu_{i \to j}(x_j) \propto \sum_{x_i \in D_i} \psi_i(x_i) \psi_{ij}(x_i, x_j) \prod_{k \in N(i) \setminus \{j\}} \mu_{k \to i}(x_i), \tag{2}$$

where $N(j)$ denotes the neighbors of node $j$ in the graph $G$. Once the algorithm has converged, the marginals are estimated as $\Pr[X_i = x_i] \propto \psi_i(x_i) \prod_{j \in N(i)} \mu_{j \to i}(x_i)$.

The update rule (2) can be applied in arbitrary order. The standard *synchronous belief propagation* updates all the message simultaneously; in each global round $t = 1, 2, 3, \ldots$, given message values $\mu_{i \to j}^t$ for all pairs $\{i, j\} \in E$, the new values $\mu_{i \to j}^{t+1}$ are computed as

$$\mu_{i \to j}^{t+1}(x_j) \propto \sum_{x_i \in D_i} \psi_i(x_i) \psi_{ij}(x_i, x_j) \prod_{k \in N(i) \setminus \{j\}} \mu_{k \to i}^t(x_i).$$

**Asynchronous belief propagation.** Starting with Elidan et al. [14], there has been a line of research arguing that *asynchronous* or *iterative* schedules for belief propagation tend to converge more reliably and with fewer message updates that the synchronous schedule. In particular, practical work has focused on schedules that attempt to iteratively perform 'the most useful' update at each step; the most prominent of these algorithms is the *residual belief propagation* of Elidan et al. [14], with other proposals aiming to address its shortcomings in various cases.

**Residual belief propagation.** Given a current state of messages, let $\mu_{i \to j}'$ denote the message we would obtain by applying the message update rule (2) to message $\mu_{i \to j}$. In residual belief propagation, the priority of a message is given by the *residual* $\mathrm{res}(\mu_{i \to j})$ of a message $\mu_{i \to j}$, defined as

$$\mathrm{res}(\mu_{i \to j}) = \left\| \mu_{i \to j}' - \mu_{i \to j} \right\|, \tag{3}$$

where $\|\cdot\|$ is an arbitrary norm; in this work, we assume $L^2$ norm is used unless otherwise specified. That is, the residual of a message corresponds to amount of change that would happen if message $\mu_{i \to j}$ would be updated. Note that this means that residual belief propagation performs *lookahead*, that is, the algorithm precomputes the future updates before applying them to the state of the algorithm. We will explore the performance of this base algorithm, as well as additional variants with *weight decay* [20] and *without lookahead* [37].

## 2.2 Parallel and Distributed Belief Propagation

The question of parallelizing belief propagation is not fully understood. The synchronous schedule is trivially parallelizable by performing updates within each round in parallel, but the improved convergence properties of iterative schedules cannot easily be translated to parallel setting. Recent proposals aim to bridge this gap in an ad-hoc manner by designing custom algorithms for specific parallel settings. We discuss the most relevant ones below, omitting ones that apply strictly to a multi-machine distributed setting (e.g. Gonzalez et al. [17].)

**Residual splash.** *Residual splash* [15] is a vertex-based algorithm inspired by residual BP. Residual splash was designed for MapReduce computation, and it aims to have larger individual tasks while retaining a similar structure to residual BP. Specifically, the algorithm works by defining a priority function over nodes of the Markov random field, and selecting the next node to process in a strict priority order. For the selected node, the algorithm performs a *splash* operation that propagates information within distance $H$ in the graph; in practice, this results in threads performing larger individual tasks at once, offsetting the cost of accessing the strict scheduler.

**Mixed synchronous and priority-based belief propagation.** Mixed strategies for belief propagation scheduling have also been proposed, by Van der Merve et al. [11] for GPUs and by Yin and Gao [40] for distributed setting. These proposals use residuals or other similar priority functions to select a set of high-score messages to update at each step, and then perform those updates as in the synchronous schedule. While these algorithms work well in their original contexts, where relaxed schedulers and other concurrent data structures cannot be utilized, this strategy is not efficient on shared-memory parallel setting on CPUs (see full version for experimental comparison.)

## 3 Relaxed Priority-based Belief Propagation

In this section, we describe our framework for parallelizing belief propagation schedules via relaxed schedulers. The main idea of the framework follows the description given in the introduction; however, we generalize it to capture schedules that do not use individual messages as elementary tasks, e.g. residual belief propagation [15].

### 3.1 Relaxed Scheduling for Iterative Algorithms

Relaxed schedulers are a basic tool to parallelize iterative algorithms, used in the context of large-scale graph processing [16, 26, 7, 12, 13]. At a high level, such iterative algorithms can be split into *tasks*, each corresponding to a local operation involving, e.g., a vertex and its edges. A standard example is Dijkstra's classic shortest-paths algorithm, where each task updates the distance between a vertex and the source, as well as the distances of the vertex's neighbours. In many algorithms, tasks have a natural priority order—in Dijkstra's, the top task corresponds to the vertex of minimum distance from the source. Many graph algorithms share this structure [36, 2], and can be mapped onto the priority-queue pattern described in the introduction. However, due to contention on the priority queue, implementing this pattern in practice can negate the benefits of parallelism [26].

**Relaxed scheduler definition.** A natural idea is to downgrade the guarantees of the perfect priority queue, to allow for more parallelism. Relaxed schedulers [1] formalize this relaxation as follows. We are given a parameter $q$, the degree of relaxation of the scheduler. The $k$-relaxed scheduler is a *sequential* object supporting `Insert (<key, priority>)`, `IncreaseKey (<key, priority>)`, with the usual semantics, and an `ApproxDeleteMin()` operations, ensuring:

(1) **Rank Bound.** `ApproxDeleteMin()` returns one of the top $k$ elements in priority order.
(2) **$q$-fairness.** A *priority inversion* on element `<key, priority>` is the event that `ApproxDeleteMin()` returns a key with a *lower* priority while `<key, priority>` is in the queue. Any element can experience at most $q$ priority inversions before it must be returned.

Relaxed schedulers are quite popular in practice, as several efficient implementations have been proposed and applied [35, 6, 38, 4, 18, 26, 31, 3, 34], with state-of-the-art results in the graph processing domain [26, 16, 19]. A parallel line of work has attempted to provide guarantees on the amount of relaxation in individual schedulers [3, 2, 33], as well as the impact of using relaxed scheduling on existing iterative algorithms [1, 5]. Here, we employ the modeling of relaxed schedulers used in e.g. [2, 5] for graph algorithms, but apply it to inference on graphical models.

### 3.2 Relaxed Priority-based Belief Propagation

Given a Markov random field, a priority-based schedule for BP is defined by a set of *tasks* $T_1, T_2, \ldots, T_K$, each corresponding to a sequence of edge updates, and a priority function $r$ that assigns a priority $r(T_i)$ to a task based on the current state of the messages as well as possible auxiliary information maintained separately. As discussed in the introduction, a non-relaxed algorithm can store all tasks in a priority queue, iteratively retrieve the highest-priority task, perform all its message updates, and update priorities accordingly.

The relaxed variant works in exactly the same way, but assuming a *$q$-relaxed* priority scheduler $Q_q$. More precisely, the following steps are repeated until a fixed convergence criterion is reached:

(1) $T_i \leftarrow Q_q.$`ApproxDeleteMin()` selects a task $T_i$ among the $q$ of highest priority in $Q_q$.
(2) Perform all message updates specified by the task $T_i$.
(3) Update the priorities for all tasks.

Note that tasks can be executed multiple times in this framework. In particular, we assume that the priority $r(T_i)$ of a task $T_i$ can only remain the same or increase when other tasks are executed, and the only point where the priority decreases is when the task is actually executed.

### 3.3 Concurrent Implementation

The sequential version of a priority-based schedule for belief propagation can be implemented using a priority queue $Q$. One could map the sequential pattern directly to a parallel setting, by replacing the sequential priority queue with a linearizable concurrent one. However, this may not be the best option, for two reasons. First, it is challenging to build *scalable* exact priority queues [21]—the data structure is inherently contended, which leads to poor cache behavior and poor performance. Second, in this context, linearizability only gives the illusion of atomicity with respect to task message updates: the data structure only ensures that the *task removal* is atomic, whereas the actual message updates which are part of the task are not usually performed atomically together with the removal.

**The multiqueue.**   For this reason, in our framework, we use a *relaxed* priority scheduler, i.e. a scalable approximate priority queue called the Multiqueue [30, 3]. As the name suggests, the Multiqueue is composed of $m$ independent *exact* priority queues. To `Insert` an element, a thread picks one of the exact priority queues uniformly at random, and inserts into it. To perform `ApproxDeleteMin()`, the thread picks *two* of the priority queues uniformly at random, and removes the *higher priority* element among their two top elements. Although very simple, this strategy has been shown to have strong probabilistic rank and fairness guarantees:

**Theorem 1** ([3, 1])**.** *A Multiqueue formed of $p \geq 3$ priority queues ensures the rank and fairness guarantees with parameter $q = O(p \log p)$, with high probability.*

**Our implementation.**   For our purposes, we assume that each thread $i$ has one or a few local concurrent priority queues, used to store pointers to BP-specific tasks (e.g. messages), which are prioritized by an algorithm-specific function, e.g. the residual values for residual BP. We store additional metadata as required by the algorithm and the graphical model. (In our experiments, we use binary heaps for these priority queues, protected by locks.) To process a new task, the thread selects two among all the priority queues uniformly at random, and accesses the task/message from the queue whose top element has higher priority. The task is marked as *in-process* so it cannot be processed concurrently by some other thread. The thread then proceeds to perform the metadata updates required by the underlying variant of belief propagation, e.g., updating the message and the priorities of messages from the corresponding node. The termination condition, e.g., the magnitude of the largest update, is checked periodically.

## 4   Dynamics of Relaxed Belief Propagation on Trees

As we will see in Section 5, the relaxed priority-based belief propagation schedules yield fast converge times on a wide variety of Markov random fields; specifically, the number of message updates is roughly the same as for the non-relaxed version, while the running times are lower. The complementary theoretical question we examine here is whether we can give analytical bounds how much extra work—in terms of additional message updates—the relaxation incurs in the worst-case.

Unfortunately, the dynamics of even synchronous belief propagation are poorly understood on general graphs, and no priority-based algorithms provide general guarantees on the convergence time. As such, we can only hope to gain some limited understanding on why relaxation retains the fast convergence properties of the exact priority-based schedules. Here, we present some theoretical evidence suggesting that as long as a schedule does not impose long dependency chains in the sequence of updates, relaxation incurs low overhead, but also that simple (non-loopy) worst-case instances exist.

**Analytical model.**   For analysis of the relaxed priority-based belief propagation, we consider the formal model introduced by [5, 2] to analyze performance of iterative algorithms under relaxed schedulers. Specifically, we model a relaxed scheduler $Q_q$ as a data structure which stores pairs corresponding to tasks and their priorities, with the operational semantics given in Section 3. In particular, there exists a parameter $q$ such that each `ApproxDeleteMin` returns one of the $q$ highest

priority tasks in $Q_q$, and if a task $T$ becomes the highest priority task in $Q_q$ at some point during the execution, then one of the next $q$ `ApproxDeleteMin` operations returns $T$. (By [3, 1], our practical implementation will satisfy these conditions with parameter $q = O(p \log p)$ w.h.p., where $p$ is the number of concurrent threads.) Other than satisfying these properties, we assume that the behavior of $Q_q$ can be adversarial, or randomized.

We model the behavior of relaxed priority-based belief propagation by investigating the number of message updates needed for convergence when the algorithm is executed *sequentially* using a relaxed scheduler $Q_q$ satisfying the above constraints. This analysis reduces to a sequential game between the algorithm, which queries $Q_q$ for tasks/messages, and the scheduler, which returns messages in possibly arbitrary fashion. One may think of the relaxed sequential execution as a form of linearization for the actual parallel execution—reference [1] formalizes this intuition. Please see the discussion at the end of this section for a practical interpretation.

**Relaxed belief propagation on trees.**   We now consider the behavior of relaxed residual belief propagation schedules on *trees with a single source*. The setting is similar to the analysis of residual splash of Gonzalez et al. [15]. Specifically, we assume that the Markov random field and the initialization of the algorithm satisfies (1) The graph $G = (V, E)$ is a tree with a specified root $r$; and (2) The factors of the Markov random field and the initial messages are such that the residuals are zero for all messages other than the outgoing messages from the root, i.e., $\text{res}(\mu_{i \to j}) = 0$ if $i \neq r$.

These conditions mean that residual belief propagation will start from the root, and propagate the messages down the trees until propagation reaches all leaves. In particular, residual belief propagation without relaxation will perform $n - 1$ message updates before convergence, updating each message away from root once. While this setting is restrictive, it does model practical instances where the MRF has tree-like structure, such as LDPC codes (see Section 5).

To characterize the dynamics on relaxed residual belief propagation on trees with a single source, we observe that the algorithm can make two types of message updates:

– Updating a message with zero residual, in which case nothing happens (*a wasted update*). This happens if the scheduler relaxes past the range of messages with non-zero residual.
– Updating a message $\mu_{i \to j}$ with non-zero residual, in which case the residual of $\mu_{i \to j}$ goes down to zero, and the messages $\mu_{j \to k}$ for the children $k$ of $j$ may change their residuals to non-zero values (*a useful update*).

It follows that each edge will get updated only once with non-zero residual. At any point of time during the execution of the algorithm, we say that the *frontier* is the set of messages with non-zero residual, and use $F(t)$ to denote the size of the frontier at time step $t$.

To see how the size of the frontier relates to the number message updates in relaxed residual belief propagation, observe that after a useful update, we have one of the following cases:

– If $F(t) \geq q$, then the next `ApproxDeleteMin()` operation to $Q_q$ will give an edge with non-zero residual, resulting in a useful update.
– If $F(t) < q$, then in the worst case we need $q$ `ApproxDeleteMin()` operations until we perform a useful update.

Our main analytic result bounds the worst-case work incurred by relaxation in two concrete cases.

**Lemma 2.** *Assume a $q$-relaxed scheduler $Q_q$ for belief propagation in the tree model defined above. The total number of updates performed by relaxed residual BP can be bounded as follows:*

– ***Good case: uniform expansion.** If the tree model has identical and non-deterministic edge factors $\psi_{ij}$ with $\psi_{ij}(x_i, x_j) \neq 0$ for all $\{i, j\}$, then the total number of updates performed by relaxed residual BP is $n + O(Hq^2)$.*
– ***Bad case: long paths.** There exists a tree instance with height $H = o(n)$ and an adversarial scheduler where relaxed residual belief propagation performs $\Omega(qn)$ message updates.*

**Discussion.**   For a more detailed analysis of the results, we refer to the full version of the paper. To interpret the results, first note that, in practice, the relaxation factor is in the order of $p$, the number of threads, and that $H$ is usually small (e.g., logarithmic) w.r.t. the total number of baseline updates $n$. Thus, in the good case, the $O(q^2 H)$ overhead can be seen as negligible: as $p$ iterations occur in parallel, the average time-to-completion should be $n/p + O(qH)$, which suggests almost

perfect parallel speedup. At the same time, our worst-case instances shows that relaxed residual BP is not a "silver bullet:" there exist tree instances where it may lead to $\Omega(qn)$ message updates, i.e. asymptotically no speedup. We discuss how to alter the priority function to mitigate these worst-case instances on trees in the full version. The next section shows experimentally that such worst-case instances are unlikely.

# 5   Evaluation

We now empirically test the performance of the relaxed priority-based algorithms, comparing it against prior work. For the experiments, we have implemented multiple priority-based algorithms and instantiated them with both exact and relaxed priority schedulers.

**Priority-based algorithms.**   We implemented several variants of sequential belief propagation, among which synchronous (round-robin), residual, weight decay, and residual without lookahead. These variants were briefly described in Section 2, and are specified in detail in the full version. For residual splash, we implemented two variants. The first is the standard splash algorithm, as given in [16]. The second is our own optimized version we refer to as *smart splash*, which only updates messages along breadth-first-search edges during a splash operation. This variant has similar convergence as the baseline residual splash algorithm, but performs fewer message updates and should be more efficient. We include the following instantiations of the algorithms in the benchmarks and compare them against the sequential residual algorithm. Please see Section 3.3 and the full version for implementation details.

**Previous algorithms.**   We ran many possible concurrent variants for the baseline algorithms, and chose the four best to we compare against our relaxed versions. For extended results, please see the full version. First, we choose the parallel version of the standard synchronous belief propagation (Synch). We omit some synchronous algorithms such as the randomized synchronous belief propagation of Van der Merve et al. [11] since they perform consistently worse. We also include the exact residual algorithms implemented via the coarse-grained priority queue, which maintains exact priority order. Specifically, here we implemented standard the residual BP algorithm (Coarse-G) and the splash algorithm of [16] (Splash) with the best value of $H$, which we found to be 10.

We also include the randomized version of splash algorithm (RS) proposed in the journal version of the paper [15] with $H = 2$, which performed best. This algorithm uses a similar idea of relaxation, but, crucially, instead of a Multiqueue scheduler, they implement a naive relaxed queue where threads randomly insert and delete into $p$ exact priority queues. While this distinction may seem small, it is known [3] that this variant does not implement a $k$-relaxed scheduler for any $k$, as its relaxation factor grows (diverges) as more and more operations are performed, and therefore corresponds to picking tasks at random to perform. As we will see in our experimental analysis, this does result in a significant difference between the number of additional (wasted) steps performed relative to a relaxed priority scheduler. Finally, we note that we are the first to implement this algorithm.

Finally, we include the bucket algorithm (Bucket) proposed in the paper [40]. At each round until convergence, this algorithm takes the $0.1 \cdot |V|$ best vertices according to the Splash metric, and updates the messages from these vertices.

**Relaxed algorithms.**   We compare the above algorithms against the algorithms we propose, i.e. relaxed versions of residual belief propagation (Relaxed Residual), weight decay belief propagation (Weight-Decay), residual without lookahead (Priority) and smart splash (Relaxed Smart Splash) with the best value $H = 2$. For all these algorithms, the scheduler is a Multiqueue with 4 priority queues per thread, as discussed in Section 3, which we found to work best (although other values behave similarly).

**Methodology.**   We run our experiments on four MRFs of moderate size: a binary tree of size $10^7$, an Ising model [14, 20] of size $10^3 \times 10^3$, a Potts [37] of size $10^3 \times 10^3$ and the decoding of $(3, 6)$-LDPC code [29] of size $3 \cdot 10^5$.

For each pair of algorithm and model, we run each experiment five times, and average the execution time and the number of performed updates on the messages. We executed on a 4-socket Intel Xeon Gold 6150 2.7 GHz server with four sockets, each with 18 cores, and 512GB of RAM. The

| Input | Prior Work | | | | | Relaxed | | | |
|---|---|---|---|---|---|---|---|---|---|
| | Synch | Coarse-G | Splash (10) | RS (2) | Bucket | Residual | Weight-Decay | Priority | Smart Splash (2) |
| Tree | 2.538x | 0.265x | 1.648x | 2.252x | 1.692x | 1.391x | 1.282x | 1.239x | 2.121x |
| Ising | 3.009x | 0.801x | 5.393x | 11.731x | 3.311x | 6.720x | 6.276x | 5.759x | 14.175x |
| Potts | — | 0.624x | 1.041x | 11.855x | — | 7.454x | 5.978x | 5.850x | 15.235x |
| LDPC | 17.735x | 1.166x | — | 5.150x | 3.044x | 13.393x | 5.615x | — | 10.519x |

Table 1: Algorithm speedups with respect to the sequential residual algorithm. Higher is better.

| Input | Prior Work | | | | | Relaxed | | | |
|---|---|---|---|---|---|---|---|---|---|
| | Synch | Coarse-G | Splash (10) | RS (2) | Bucket | Residual | Weight-Decay | Priority | Smart Splash (2) |
| Tree | 48.000x | 1.003x | 16.442x | 8.344x | 5.110x | 1.020x | 1.012x | 3.657x | 2.565x |
| Ising | 45.006x | 1.003x | 9.266x | 5.787x | 3.996x | 1.058x | 1.068x | 1.816x | 1.878 |
| Potts | — | 1.006x | 9.005x | 5.983x | — | 1.068x | 1.053x | 1.791x | 1.891x |
| LDPC | 4.404x | 1.003x | — | 4.089x | 1.668x | 1.007x | 0.883x | — | 0.973x |

Table 2: Total updates relative to the sequential residual algorithm at 70 threads. Lower is better.

code is written in Java; we use Java 11.0.5 and OpenJDK VM 11.0.5. Our code is available at https://github.com/IST-DASLab/relaxed-bp. Our code is fairly well optimized—in sequential executions it outperforms the C++-based open-source framework of libDAI [25] by more than 10x, and by more that 100x with multi-threading. The sizes of the inputs described in the previous paragraph are chosen such that their execution is fairly long while the data still can fit into RAM.

We run the baseline algorithm on one process since it is sequential, while all other algorithms are concurrent and, thus, are executed using 70 threads. The execution times (speedups) relative to the sequential baseline are presented in Table 1. Each cell of the table shows how much faster the corresponding algorithm works in comparison to the sequential residual one, i.e., higher is better; "—" means that the execution did not converge within a reasonable limit of time.

To verify the accuracy of inference, we confirmed that all algorithms were able to recover the known correct codeword used to generate the LDPC code instance as expected. On the Ising model, all algorithms converged to almost identical marginals; on the Potts model, some non-relaxed algorithms had larger differences due to poor convergence.

**Results.** See Table 1 for the speedups versus the baseline, on 70 threads. (For ablation studies, see the full version.) On trees, the fastest algorithm is, predictably, the synchronous one, since on tree-like models with small diameter $D$ it performs only approximately $O(D)$ times more updates in comparison to the sequential baseline, while being almost perfectly parallelizable. So, it works well on perfect binary tree as our Tree model, but works much worse on the chain graphs. On Ising and Potts models, the best algorithm is Relaxed Smart Splash (RSS) with $H = 2$. The algorithm closest to it is Random Splash with $H = 2$, which is $20 - 30\%$ slower. For LDPC decoding, which is a tree-like model, the best-performing is again the Synchronous algorithm. We note the good performance of the relaxed residual algorithm, as well as of RSS, and the relatively poor performance of Random Splash, due to high numbers of wasted updates. Examining Table 2, we notice in general the relatively low number of wasted updates for relaxed algorithms. In summary, the choice of algorithm can depend on the model; however, one may choose Relaxed Smart Splash since it performs well on all our models.

# 6 Discussion

We have investigated the use of relaxed schedulers in the context of the classic belief propagation algorithm for inference on graphical model, and have shown that this approach leads to an efficient family of algorithms, which improve upon the previous state-of-the-art non-relaxed parallelization approaches in our experiments. Overall, our relaxed implementations, either Relaxed Residual or Relaxed Smart Splash, have state-of-the-art performance in multithreaded regimes, making them a good generic choice for any belief propagation task.

For future work, we highlight two possible directions. First is to extend our theoretical analysis to cover more types of instances; however, as we have seen, the structure of belief propagation schedules can quite complicated, and the challenge is the figure out a proper framework for more general analysis. Second possible direction is extending our empirical study to a massively-parallel, multi-machine setting.

## Broader Impact

As this work is focused on speeding up existing inference techniques and does not focus on a specific application, the main benefit is enabling belief propagation applications to process data sets more efficiently, or enable use of larger data sets. We do not expect direct negative societal consequences to follow from our work, though we note that as with all heuristic machine learning techniques, there is an inherent risk of misinterpreting the results or ignoring biases in the data if proper care is not taken in application of the methods. However, such risks exist regardless of our work.

## Acknowledgments and Disclosure of Funding

We thank Marco Mondelli for discussions related to LDPC decoding, and Giorgi Nadiradze for discussions on analysis of relaxed schedulers. This project has received funding from the European Research Council (ERC) under the European Union's Horizon 2020 research and innovation programme (grant agreement No 805223 ScaleML) and from Government of Russian Federation (Grant 08-08).

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
