[Supplementary Material]

# Relaxed Scheduling for Scalable Belief Propagation

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

97     work.
98   – We implement and experimentally compare different variants of belief propagation under relaxed
99     scheduling. We identify a new family of relaxed schedulers which consistently matches or
100    outperforms previous proposals. Benchmarks show that this framework gives state-of-the-art
101    parallel scalability on a wide variety of Markov random field models.

## 2   Preliminaries and related work

### 2.1   Belief Propagation

We consider marginalization in *pairwise Markov random fields*; one can equivalently consider factor
graphs or Bayesian networks [40]. A pairwise Markov random field is defined by a set of random
variables $X_1, X_2, \ldots, X_n$, a graph $G = (V, E)$ with $V = \{1, 2, \ldots, n\}$, and a set of *factors*

$$
\begin{aligned}
\psi_i &: D_i \to \mathbb{R}^+ &&\text{for } i \in V, \\

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

**Randomized synchronous belief propagation.** Van der Merve et al. [11] proposed a parallelization scheme for belief propagation on GPUs, mixing the structure of synchronous and residual belief propagation. Their algorithm considers all messages at once in global rounds, and performs *filter-and-select* steps. First, it filters out all messages whose residuals are below the convergence threshold. Second, out of the remaining messages, select a $p$-fraction uniformly at random to update. The fraction $p$ is adjusted on the fly based on the convergence of the algorithm, preferring a low value if the algorithm is converging slowly, and a high value if it is converging fast. This algorithm is well-suited for GPUs, as the filter-and-select steps can be efficiently implemented. However, as shown by our experimental study, this strategy is not efficient on CPUs, on real-world models. Conversely, as discussed in [11], the dynamic priority-based strategy we propose would be hard to implement efficiently on GPUs, due to its irregular structure.

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

*Proof.* **Good case: uniform expansion.**   As the first case, we consider the tree model in the case where the edge factors $\psi_{ij}$ are identical for all edges and not *deterministic*, i.e. $\psi_{ij}(x_i, x_j) \neq 0$ for all $\{i, j\}$. Let us say that the *level* of a message $\mu_{i \to j}$ is $\ell$ if the distance from $i$ to the root $r$ is $\ell$. The conditions we imposed our Markov random field, together with the update rule (2), imply that the residuals of the messages are decreasing in the level $\ell$ of the message, and all messages on level $\ell$ will have the same residual when they are in the frontier. This means that residual schedule will prefer updating messages on lower levels first.

Now consider the progress of the relaxed residual belief propagation on this tree; let $H$ denote the height of the tree. Now assume that all messages on levels $0, 1, \dots, \ell - 1$ have had a useful update, and consider how many wasted updates we can make in the worst case before all messages on level $\ell$ have been processed. Let $f$ denote the number of message on level $\ell$ still in the frontier:

– While $f \geq k$, there are at least $k$ messages of level $\ell$ on the frontier. Since they have the highest residual out of the messages in the frontier, each update is a useful update of a message on level $\ell$.
– When $f < k$, there can be updates that do not hit messages on level $\ell$, which can possibly be wasted updates. However, the highest-priority messages are still from level $\ell$, so every $k$th update will hit a message on level $\ell$ by the guarantees of the scheduler. Thus, in $(k - 1)f = O(k^2)$ updates, all remaining messages on level $\ell$ have been processed.

Since there can be at most $n - 1$ useful updates, and the number of levels is $H - 1$, the total number of updates performed by relaxed residual belief propagation is $n + O(Hk^2)$.

**Bad case: long paths.**   A simple example where relaxed residual belief propagation performs poorly is a path. That is, if our underlying tree is a path of length $n$ with a root at one end, then relaxed residual belief propagation can perform $\Omega(kn)$ message updates in the worst case. However, the path has height $H = n$, so one might ask if there is a general upper bound of form $n + O(Hk^2)$ on trees without restricting the edge factors as in our previous example.

Unfortunately, turns out that without the restrictions above, we can construct examples of trees with height $H = o(n)$ where relaxed residual belief propagation still performs $\Omega(kn)$ message updates (see Figure 2 for an illustration):

(1)  Start with a path of length $\sqrt{n}$, with a root at one end.
(2)  Attach a new path of length $\sqrt{n}$ to each vertex.
(3)  For each remaining degree-2 node in the graph, attach a single new node to it.

This construction results in a 3-regular rooted tree with $\Theta(n)$ nodes and depth $H = O(\sqrt{n})$. Finally, we choose the edge factors so that residuals on the side paths are larger than the residuals on the main path, so residual belief propagation will prefer following the side paths first.

One can now observe that under the adversarial model for the relaxed scheduler, the adversary can select the execution of the relaxed scheduler so that the frontier size never exceeds 4. That is, adversary forces the algorithm to process the graph one side path at time, wasting $k - 1$ steps between each useful update.

Figure 2: Example of the tree where relaxed residual belief propagation performs poorly.

Finally, we note that the same construction can be generalized to obtain instances with similar relaxation overhead and diameter $O(n^{1/c})$ for larger constants $c < k$, by simply working with paths of length $n^{1/c}$ and repeating the path attachment step $c$ times.

$\square$

*Remark* 3. As suggested by the above examples, one might consider changing the priority function to preferentially select messages closer to the source. This can lead to improved work guarantees for the relaxed schedule. Indeed, we discuss one concrete example in Section B, where we show how to relax the optimal schedule on trees. However, it is not straightforward to construct such priority functions so that they also make sense on general graphs, which can have non-monotonic potentials and cycles.

# B   Optimal schedule on trees

On trees, the belief propagation gives exact marginals under any schedule that updates each edge infinitely often. However, there is an optimal schedule that updates each message exactly once, requiring $O(n)$ message updates [27]. Assume the tree has a fixed root $v$:

(1) In the first phase, all messages towards the root are updated starting from the leaves; each message is updated only after all its predecessors have been updated.

(2) In the second phase, all messages away from the root are update starting from the root.

This schedule can be modeled in the priority-based scheduling framework as follows:

(1) Initially, the outgoing messages at leaf nodes have priority $n$, and all other messages have priority 0.

(2) When message is updated with non-zero priority, its priority is changed to 0.

(3) Once all messages $\mu_{k \to i}$ for $k \in N(i) \setminus \{j\}$ have been updated once with non-zero priority, the message $\mu_{i \to j}$ changes to priority to minimum of update priorities of the incoming edges minus one.

This priority function can clearly be implemented by keeping a constant amount of extra information per message. When the above schedule is executed with an exact scheduler, the algorithm will update each message once with non-zero priority before considering any messages with zero priority, and by following the analysis of [27], one can see that the algorithm has converged at that point.

Similarly, in the relaxed version of the schedule, the algorithm has converged once all messages have been updated once with non-zero priority. In addition, some messages may be updated multiple times with priority 0; we call these *wasted* updates, and the updates done while the message has non-zero priority *useful* updates.

**Claim 4.** *The relaxed version of the optimal schedule on trees performs $O(n + k^2 H)$ message updates, where $H$ is the height of the tree.*

*Proof.* For the purposes of analysis, assign messages into buckets $B_1, B_2 \ldots, B_n$ so that bucket $B_\ell$ contains the messages that will have their useful update done with priority $\ell$. One can observe that the update priority of message $\mu_{i \to j}$ is the $n - d$, where $d$ is the maximum distance from node $i$ to a leaf using a path that does not cross edge $\{i, j\}$. Since this is bounded by the diameter of the tree, there are at most $2H$ non-empty buckets.

Assume that all messages in buckets $B_n, B_{n-1}, \ldots, B_{\ell+1}$ have been already had a useful update. We now show that in there can be at most $k^2$ wasted updates before all messages in $B_\ell$ have had a useful update. Since all earlier buckets have been processed, all messages in $B_\ell$ have either already had a useful update, or have priority $\ell$. Let $b$ be the number of messages remaining in bucket $B_\ell$:

– While $b \geq k$, there are at least $k$ messages with priority $\ell$, so each update is a useful update of a message in $B_\ell$

– When $b < k$, there can be wasted updates. However, since buckets $B_n, B_{n-1}, \ldots, B_{\ell+1}$ have had all useful updates, the top elements in the schedule will be from bucket $B_\ell$, and

thus by the guarantees of the scheduler, there can be at most $k - 1$ wasted updates before the top element is processed. Thus, in $b(k-1) = O(k^2)$ updates, all remaining messages of $B_\ell$ will have their useful update.

By an inductive argument, all non-empty buckets have been processed after $O(k^2 H)$ wasted updates, so the total number of updates is $O(n + k^2 H)$. $\qquad\qquad\square$

## C  Algorithms

### C.1  Asynchronous belief propagation

Starting with Elidan et al. [14], there has been a line of research arguing that *asynchronous* or *iterative* schedules for belief propagation tend to converge more reliably and with fewer message updates that the synchronous schedule. In particular, the practical work has focused on developing schedules that attempt to iteratively perform 'the most useful' update at each step; the most prominent of these algorithms is the *residual belief propagation* of Elidan et al. [14], with other proposals aiming to address the shortcomings of residual belief propagation in various cases.

**Residual belief propagation.** Given a current state of messages, let $\mu'_{i \to j}$ denote the message we would obtain by applying the message update rule (2) to message $\mu_{i \to j}$. In residual belief propagation, the priority of a message is given by the *residual* $\mathrm{res}(\mu_{i \to j})$ of a message $\mu_{i \to j}$, defined as

$$\mathrm{res}(\mu_{i \to j}) = \left\| \mu'_{i \to j} - \mu_{i \to j} \right\|, \qquad\qquad (4)$$

where $\|\cdot\|$ is an arbitrary norm; in this work, we assume $L^2$ norm is used unless otherwise specified. That is, the residual of a message corresponds to amount of change that would happen if message $\mu_{i \to j}$ would be updated. Note that this means that residual belief propagation performs *lookahead*, that is, the algorithm precomputes the future updates before applying them to the state of the algorithm.

**Weight decay belief propagation.** *Weight decay belief propagation* of [20] is a variant of residual belief propagation that penalizes message priorities for repeated updates. That is, let $m(\mu_{i \to j})$ denote how many times message $\mu_{i \to j}$ has been updated by the algorithm, and let $\mathrm{res}(\mu_{i \to j})$ denote the residual of a message as above. The priority function of weight decay belief propagation is

$$r(\mu_{i \to j}) = \frac{\mathrm{res}(\mu_{i \to j})}{m(\mu_{i \to j})} .$$

The motivation behind this weight decay scheme is that empirical observations suggest that one possible failure mode of residual belief propagation is getting stuck in cycles with large residuals; the weight decay prioritizes other edges in cases where this happens.

**Residual without lookahead.** Another variant of residual belief propagation is the lookahead-avoiding belief propagation of [38]. As the name implies, this algorithm does not perform the exact residual computation using (4), but instead approximates the residuals indirectly, with the aim of reducing the computational cost of priority updates.

Informally, the basic idea is that for each message $\mu_{i \to j}$, we track the amount other incoming messages at node $i$ have changed since the last update of $\mu_{i \to j}$, and use this to define the priority of updating $\mu_{i \to j}$. The actual approximation in the algorithm uses a slightly different notion of residual from (4), so we refer to [38] for full details.

### C.2  Parallel belief propagation

As discussed above, the question of parallelizing belief propagation is fairly poorly understood. The synchronous schedule is trivially parallelizable by performing updates within each round in parallel, but the improved converge properties of the iterative schedules cannot easily be translated to parallel setting. There have been recent proposals that aim to bridge this gap in an ad-hoc manner by designing custom algorithms for specific parallel computation settings.

**Residual splash.** The *residual splash* belief propagation [16] is a vertex-based algorithm inspired by residual belief propagation. The residual splash algorithm was initially designed for MapReduce computation, and it aims to have larger individual tasks while retaining a similar structure to residual belief propagation.

Specifically, the residual splash algorithm works by defining a priority function over nodes of the Markov random field, and selecting the next node to process in a strict priority order. For the selected node, the algorithm performs a *splash* operation that propagates information within distance $H$ in the graph; in practice, this results in threads performing larger individual tasks at once, offsetting the cost of accessing the strict scheduler.

In detail, the priority of for nodes is given by the *node residual*, defined as

$$\text{res}(i) = \max_{j \in N(i)} \text{res}(\mu_{j \to i}).$$

Given a *depth parameter* $H$, the splash operation at node $i$ is defined by following sequence of message updates:

(1) Construct a BFS tree $T$ of depth $H$ rooted at node $i$.

(2) In the reverse BFS order on $T$—starting from leaves—process all nodes in $T$, updating all outgoing messages for each node processed.

(3) Repeat the previous step in BFS order, i.e., starting from the root.

In other words, this process gather all available information at radius $H$ from the selected node, and propagates it to all nodes within the radius.

**Randomized synchronous belief propagation.** Van der Merve et al. [11] proposed a parallelization scheme for belief propagation on GPUs, mixing the structure of synchronous and residual belief propagation. Their algorithm considers all messages at once in global rounds, and performs the following filter-and-select steps before computing the message updates:

(1) Filter out all messages whose residuals are below the convergence threshold.

(2) Out of the remaining messages, select a $p$ fraction of messages uniformly at random to update.

Alternatively, the process can perform the algorithm on per-node basis, using node residuals as in the residual splash algorithm.

The fraction $p$ is adjusted on the fly based on the convergence of the algorithm, preferring a low value if the algorithm is converging slowly, and a high value if it is converging fast. Concretely, the selection scheme for $p$ used by [11] is to set $p = 1$ if the number of messages above the convergence threshold decreased by at least $10\%$ in the last round, and set it to a smaller fixed value otherwise.

We note that the randomized synchronous algorithm is particularly well suited for GPU use, as the filter-and select steps can be efficiently implemented on GPUs. However, as shown by our experimental study, this strategy is not efficient on a subset of real-world models, when ported to CPU. Conversely, as discussed by the authors of [11], the dynamic priority-based strategy we propose would be hard to implement efficiently on GPUs, due to its irregular structure.

# D  Models

We run our experiments on four Markov random fields models.

**Trees.** As a simple base case, we consider a simple tree model similar to the analytical setting in Section 4. The underlying graph is a full binary tree on 10 millions vertices, and the other parameters are set up as follows:

– All variables are binary, i.e. the domain is $\{0, 1\}$ for each variable.

– Vertex factors are $(0.1, 0.9)$ for the root and $(0.5, 0.5)$ for all other vertices.

– Edge factors are $\psi_{ij}(x, y) = \begin{cases} 1, & x = y \\ 0, & x \neq y \end{cases}$ for all edges.

As discussed in Section 4, these choices create a setup where the belief propagation has to propagate information from the root to all other nodes. Thus, under an optimal schedule, the total number of performed updates is be equal to $10^7 - 1$. Since we know that all algorithms will converge on this model, we run the algorithms until exact convergence.

**Ising and Potts models.** Ising and Potts models are Markov random fields defined over an $n \times n$ grid graph, arising from applications in statistical physics. Both of Ising [14, 20] and Potts [38] models were used in prior work as test case, and in general they offer a class of good test instances, as they both exhibit complex cyclic propagations and are easy to generate.

For the parameters of the models, we mostly follow prior work in the setup. As the underlying graph, we use a $10^3 \times 10^3$ grid graph to get instances where the effects of parallelization are clearly visible. For the Ising model, we select the factors similarly to [14, 20]:

- The variable domain is $\{-1, 1\}$ for all variables.
- Vertex factors are $\psi_i(x) = e^{\beta_i x}$.
- Edge factors are $\psi_{ij}(x, y) = e^{\alpha_{ij} xy}$.
- The parameters $\alpha_{ij}$ and $\beta_i$ are chosen uniformly at random from $[-1, 1]$.

For the Potts model, we select the factors following [38]:

- The variable domain is $\{0, 1\}$ for all variables.
- Vertex factors are $\psi_i(x) = \begin{cases} e^{\beta_i}, & x = 1 \\ 1, & x = 0 \end{cases}$.
- Edge factors are $\psi_{ij}(x, y) = \begin{cases} e^{\alpha_{ij}}, & x = y \\ 1, & x \neq y \end{cases}$.
- The parameters $\alpha_{ij}$ and $\beta_i$ are chosen uniformly at random from $[-2.5, 2.5]$.

For both Ising and Potts models, we set the convergence threshold to $10^{-5}$. That is, we terminate algorithm once all task have priority below this threshold.

**LDPC codes.** Finally, we generate Markov random fields corresponding to the $(3, 6)$-LDPC (*low density parity check code* [15]) decoding. LDPC decoding is one of the more successful application of belief propagation. We consider a simple version of LDPC decoding task where convergence guarantees exist [29]. However, we stress that coding theory is its own extensive research area, and far more optimized codes and decoding algorithms exist in practice—we simply use LDPC decoding to observe the comparative scaling behavior of our implementations on instances where synchronous belief propagation is guaranteed to converge. For a more detailed background on LDPC decoding and other aspects of coding theory, refer e.g. to the book [30].

More precisely, we consider $(3, 6)$-LDPC decoding over a binary symmetric channels. Informally, a $(3, 6)$-LDPC code is a $(3, 6)$-regular bipartite graph, where each degree 3 node corresponds to a binary *variable* and each degree 6 node corresponds to a *constraint* of form $x_{i_1} + x_{i_2} + \ldots + x_{i_6} = 0$ over the neighboring variables $x_{i_1}, x_{i_2}, \ldots, x_{i_6}$. Each sequence of variables that satisfies the all the constraints is *codeword* of the code. The basic setup is then that we send a codeword over a *channel* that flips each bit with probability $\varepsilon$, and the receiver will run belief propagation and use results of marginalization to infer the original codeword.

For our experiments, we build a $(3, 6)$-LDPC instance with $300\,000$ variable nodes and $150\,000$ constraint nodes by selecting a random $(3, 6)$-regular bipartite graph, and initialize the node factors corresponding to the all-zero codeword sent over binary symmetric channel with error probability $\varepsilon = 0.07$. Under these conditions, belief propagation is guaranteed to correctly decode the instance with high probability [29]; indeed, all the algorithms that converged decoded the codeword correctly in our experiments. The codeword length was again selected to get roughly comparable baseline running times as for the other instances.

Concretely, we get Markov random field where the underlying graph is a random bipartite graph with $450\,000$ nodes. For each variable node $i$, let $x_i \in \{0, 1\}$ be the 'transmitted' value of the variable, randomly generated to be 1 with probability $\varepsilon$ and 0 otherwise. The factors have the following structure:

- The domains of variable nodes are binary domains $\{0, 1\}$. For the constraint nodes, the domain is $\{0, 1\}^6$—different bit masks of length 6.
- The node factors for variable nodes are

$$\psi_i(y) = \begin{cases} 1 - \epsilon, & y = x_i \\ \epsilon, & y \neq x_i. \end{cases}$$

| Input | Residual | Prior Work | | | | | | Relaxed | | | | |
|---|---|---|---|---|---|---|---|---|---|---|---|---|
| | | Synch | CG | S 2 | S 10 | RS 2 | RS 10 | Residual | WD | Priority | RSS 2 | RSS 10 |
| Tree | 1.30 min | 2.538x | 0.265x | 0.608x | 1.648x | 2.252x | 2.241x | 1.391x | 1.282x | 1.239x | 2.121x | 2.110x |
| Ising | 2.76 min | 3.009x | 0.801x | 0.609x | 5.393x | 11.731x | 13.512x | 6.720x | 6.276x | 5.759x | 14.175x | 10.337x |
| Potts | 3.02 min | — | 0.624x | 0.484x | 1.041x | 11.855x | 12.854x | 7.454x | 5.978x | 5.850x | 15.235x | 11.091x |
| LDPC | 4.62 min | 17.735x | 1.166x | — | — | 5.150x | — | 13.393x | 5.615x | — | 10.519x | — |

Table 3: Algorithm speedups with respect to the sequential residual algorithm. Higher is better.

| Input | Residual | Prior Work | | | | | | Relaxed | | | | |
|---|---|---|---|---|---|---|---|---|---|---|---|---|
| | | Synch | CG | S 2 | S 10 | RS 2 | RS 10 | Residual | WD | Priority | RSS 2 | RSS 10 |
| Tree | 10M | 48.000x | 1.003x | 8.658x | 16.442x | 8.344x | 15.197x | 1.020x | 1.012x | 3.657x | 2.565x | 5.027x |
| Ising | 25.3M | 45.006x | 1.003x | 5.719x | 9.266x | 5.787x | 10.232x | 1.058x | 1.068x | 1.816x | 1.878x | 6.147x |
| Potts | 30M | — | 1.006x | 5.903x | 9.005x | 5.983x | 9.109x | 1.068x | 1.053x | 1.791x | 1.891x | 6.328x |
| LDPC | 7.23M | 4.404x | 1.003x | — | — | 4.089x | — | 1.007x | 0.883x | — | 0.973x | — |

Table 4: Total updates relative to the sequential residual algorithm at 70 threads. Lower is better.

For the constraint nodes, the node factor $\psi_c(y)$ is equal to the number of ones in $y \in \{0,1\}^6$ modulo 2; this effectively penalizes any value that does not satisfy the constraint.

– Edge factors $\psi_{ic}(x,y)$ is one if the corresponding bit in the $y \in \{0,1\}^6$ equals $x \in \{0,1\}$, and is zero otherwise.

For the LDPC instances, we set the convergence threshold to $10^{-2}$ to ensure fast convergence; this approximates the behavior of actual LDPC decoders.

# E   Experiments

In this section, we provide an additional study on the evaluation of the algorithms. At first, we give the extended results of running the algorithms on the moderate size inputs chosen in the main body of the paper. Unfortunately, due to the reasonably high running time it was impossible to make enough points for the plots to reason about the general effect of the parallelization. Thus, we execute the algorithms on a little bit smaller inputs.

## E.1   Moderate size inputs

To present all the executed algorithms in the table, we shrink the abbreviations a little bit: Coarse-Grained now becomes CG, Splash becomes S, Random Splash becomes RS, Relaxed Residual rests Residual, Weight-Decay becomes WD, Relaxed Priority becomes Priority, and, finally, Relaxed Smart Splash becomes RSS. Table 3 contains the execution times (speedups) of the algorithms relative to the sequential baseline. Table 4 contains the number of updates performed by the algorithms in compare to the number of updates performed by the sequential baseline. The results do not differ much from the ones presented in the main body of the paper. The only notable thing is that Random Splash with $H = 10$ it performs better on Ising and Potts model than Random Splash with $H = 2$. However, we chose Random Splash with $H = 2$ as the best one, since Random Splash with $H = 10$ does not finish on LDPC input. Nevertheless, `Relaxed Smart Splash` outperforms Random Splash with both settings of $H$.

## E.2   Small size inputs

In this subsection, we decrease the size of the inputs. Now, Tree model maintains a tree of size $10^6$, Ising and Potts models are built on top of $300 \times 300$ grid graph, and, finally, LDPC model is set up with $30\,000$ length of the input vector. In general, we simply reduce the sizes of the models by approximately 10.

### E.2.1   Scaling

**How to read the plots.**   There are two types of plots per each model: the first shows the execution time of the algorithms, while the other one shows the number of updates performed. On the $x$ axis we have the number of threads the algorithms were run on, while on the $y$ axis we have: the time in seconds (for time plots) and the number of updates (for update plots). The dashed lines on the plots correspond to the algorithms that use a relaxed scheduler, while the others use either no concurrent scheduler, or an exact priority queue.

(a) Execution time      (b) Number of updates

Figure 3: The results of the evaluation of the algorithms on the Tree model

(a) Execution time      (b) Number of updates

Figure 4: The results of the evaluation of the algorithms on Ising model

Whenever we have omitted algorithms from the plots or display incomplete data, this indicates poor performance for that algorithm on the metric displayed on the graph: either the algorithm did not converge or the values exceed the limit of the plot.

**Tree model.** As one can observe on the time plot (Figure 3a), the three algorithms with the best scaling on the tree instance are the synchronous belief propagation, relaxed residual and the weight-decay algorithm. For the relaxed algorithms, this mirrors our theoretical analysis from Section 4: as can be seen from Figure 3b, the relaxation incurs very low overhead in terms of additional updates, while the overhead from parallelization is also low. By contrast, the exact residual belief propagation performs exactly the minimum number of updates needed, but scales very badly due to the contention on the priority queue.

We note that on the tree instance, the synchronous belief propagation also scales very well when parallelized. The amount of work can be split evenly between the threads, and only $O(\log n)$ synchronous rounds are required for convergence.

**Ising and Potts model.** Ising and Potts models represent more challenging instances with lots of cycles, and are generally thought to be more representative of hard general graph instances for belief propagation. As can be seen in Figures 4a and 5a, relaxed algorithms perform consistently well on these instances, with relaxed residual belief propagation giving consistently the fastest convergence. These are followed by the exact splash algorithms, which generally perform slightly worse; however,

(a) Execution time

(b) Number of updates

Figure 5: The results of the evaluation of the algorithms on Potts model

(a) Execution time

(b) Number of updates

Figure 6: The results of the evaluation of the algorithms on decoding LDPC code

the scaling seems to be somewhat sensitive to the choice of the parameter $H$. Both the synchronous and exact residual belief propagation are omitted, as the former did not consistently converge, and the latter was very slow.

An interesting insight is that the exact variants of splash and smart splash do not converge at all in single-threaded executions for some values of the parameter $H$, but always converge on two and more threads. Similarly, synchronous belief propagation, which has a fixed schedule, does not converge. By contrast, relaxed smart splash converged under all parameter values. We conjecture that this is due to the phenomenon observed by [20]: exact priority-based algorithms may get stuck in non-convergent cyclic schedules, and injecting randomness into the schedule may help the algorithm to 'escape' these situations. In particular, relaxation to the priority queue, i.e., sometimes executing low-priority items, can provide a such source of randomness. Similarly, an increase in the number of threads leads to the relaxation of the algorithm even for exact schedulers, as several messages are processed in parallel, not only the best one. Thus, we empirically observe that the randomness in the relaxation might help belief propagation to avoid bad cyclic schedules and, therefore, converge.

**LDPC model.** There are five algorithms that perform similarly (Figure 6a): synchronous belief propagation, relaxed residual belief propagation, the weight decay algorithm, relaxed smart splash with $H = 2$ and, finally, smart splash with $H = 2$. The other algorithms did not converge within our five minutes time limit per experiment.

|          | Threads | Message updates | | | |
|----------|---------|---------|---------|---------|---------|
|          |         | Tree | Ising | Potts | LDPC |
| Exact    | 1       | 1000000 | 2279000 | 2700000 | 1464000 |
| Relaxed  | 1       | +0.14% | +0.11% | -0.01% | +0.55% |
|          | 2       | +0.26% | +0.24% | +0.37% | +0.57% |
|          | 6       | +0.56% | +2.50% | +2.70% | +0.64% |
|          | 10      | +0.92% | +3.71% | +4.45% | +1.05% |
|          | 20      | +2.08% | +5.27% | +5.87% | +1.41% |
|          | 30      | +2.90% | +6.10% | +6.56% | +1.87% |
|          | 40      | +3.48% | +6.52% | +7.39% | +2.35% |
|          | 50      | +5.04% | +6.83% | +7.92% | +2.83% |
|          | 60      | +4.96% | +7.39% | +8.28% | +3.20% |
|          | 70      | +5.74% | +7.71% | +8.53% | +3.70% |

Table 5: Number of additional message updates performed by relaxed residual belief propagation compared to exact residual belief propagation.

We note that synchronous belief propagation performs very well on this instance. This is not surprising, as standard belief propagation is known to perform well in LDPC decoding. Generally speaking, the necessary propagation chains seem to be very short on LDPC instances, and the synchronous algorithm parallelizes well in such cases.

### E.2.2 The effects of relaxation

In Table 5, we measure how many more updates the relaxed residual algorithm needs to perform in comparison to the number of updates performed by the standard sequential residual algorithm, denoted as "baseline". We count the total number of updates only approximately: we check the convergence condition only after every 1000 iterations.

The left column indicates whether it is a baseline algorithm or the number of threads for relaxed residual belief propagation. The other columns present the numbers for each model we consider. Each cell contains the corresponding number of updates and how many more updates the relaxed version of the algorithm executed (percentage).

On one process, relaxed residual performs more updates than the baseline does, except in the case of the Potts model. It is expected since our algorithm uses relaxed Multiqueue instead of the strict priority queue. Moreover, as expected the overhead on the number of updates in comparison to the baseline increases with the number of threads. This is again due to the relaxation of the priority queue–recall that we allocate $4\times$ more queues than threads. Interestingly, this overhead is limited even on 70 threads—its maximum value is $9\%$ maximum. This explains the good performance of our algorithm: we reduce the contention by relaxing accesses to the priority queue, while at the same time the total number of updates does not increase significantly.

### E.2.3 Relaxed versus Non-Relaxed Algorithms

In Table 6, we analyze the speedups obtained by the relaxed residual algorithm relative to the best-performing non-relaxed alternative across models and thread counts. We notice that our algorithm outperforms the alternatives in most of the cases, often by a large margin—the highest speedup is of $2.85\times$, whereas the highest slow-down is of $0.47x$. Both occur on the Potts model, which is generally the most difficult instance in our tests. Overall, the combination of our relaxed scheduling framework combined with the standard residual belief propagation is clearly the algorithm of choice at high thread counts, where it consistently outperforms the alternatives; on the other hand, relaxed residual also performs reasonably well on a single thread, making it a consistently good choice all across the board.

### E.2.4 Random Synchronous Algorithm

In Table 7, we present the execution time of random synchronous algorithm on 70 threads (Random Synch 70) with different values of $lowP = 0.1, 0.4$ and $0.7$, where the parameter $lowP$ controls the

| Threads | Speedup | | | |
|---|---|---|---|---|
| | Tree | Ising | Potts | LDPC |
| 1 | 0.89x | **1.08x** | **1.04x** | **1.14x** |
| 2 | 0.75x | 0.51x | 0.47x | **1.13x** |
| 6 | **1.20x** | 0.77x | 0.73x | **1.17x** |
| 10 | **1.16x** | **1.01x** | 0.94x | **1.20x** |
| 20 | **1.36x** | **1.66x** | **1.89x** | **1.49x** |
| 30 | **1.38x** | **1.88x** | **1.82x** | **1.65x** |
| 40 | **1.61x** | **2.21x** | **1.90x** | **1.62x** |
| 50 | **1.91x** | **2.67x** | **2.36x** | **1.48x** |
| 60 | **1.89x** | **2.66x** | **2.85x** | **1.55x** |
| 70 | **1.61x** | **2.71x** | **2.44x** | **1.52x** |

Table 6: Speedup of relaxed residual belief propagation versus the best non-relaxed alternative on different thread counts. We note that overhead of parallelization can overcome the benefits on small thread counts, as seen in the scaling experiments.

random selection fraction $p$ in steps where the algorithm is converging slowly (see Section C.2). We compare it with the execution time of two baselines: Synchronous algorithm on 70 threads (Synch 70) and Relaxed Residual on one process (RR 1). Cells with '—' indicate executions that either take more than five minutes to run or simply do not converge.

To summarize, we did not include the execution time of random synchronous algorithm in the scaling plots since it exceeds the execution time of one of the baselines in all cases.

| Algorithm | | Running time (s) | | | |
|---|---|---|---|---|---|
| | | Tree | Ising | Potts | LDPC |
| Synch 70 | | 4.088 | — | — | 3.504 |
| RR 1 | | 5.579 | 9.012 | 10.583 | 25.663 |
| Random Synch 70 | $lowP = 0.1$ | 37.052 | 62.629 | — | 28.543 |
| | $lowP = 0.4$ | 8.420 | 20.396 | — | 7.269 |
| | $lowP = 0.7$ | 6.306 | 12.581 | — | 4.791 |

Table 7: Randomized synchronous algorithm versus baselines.