[Reviews · NeurIPS 2020]

Review 1

Summary and Contributions: The authors describe a practical contribution to parallel belief propagation implementations, by applying existing parallel algorithms for "relaxed scheduling" that are popular in parallel graph processing to priority-based message passing inference. The relaxed scheduler returns a near-top element using a data structure that leads to less resource contention among the concurrent processes.

Strengths: The paper is very practical in focus, with excellent experimental evaluations that highlight the strengths of the method (generally robust speedups across problem types). The method is quite simple in concept and builds on existing priority-based parallel scheduling work combined with a useful technique from other graph processing literature. The work makes a clear contribution to the several existing papers on parallel BP implementations. They also make a minor modification to Gonzalez et al. (2009)'s splash ("smart splash") that appears to improve performance.

Weaknesses: Novelty level is medium -- the work is a combination of existing ideas (splash, relaxed scheduling) that have previously appeared. The combination is not surprising, but to my knowledge is new. The authors provide some basic theoretical analysis of their method, but these are fairly trivial in nature (worst-case number of messages on tree graphs, etc.)

Correctness: Yes

Clarity: Yes, very well written.

Relation to Prior Work: Yes, very clearly. The differences with some existing work are sometimes small but important (e.g., the relationship to "random splash") and so the clarity is very helpful.

Reproducibility: Yes

Additional Feedback:


Review 2

Summary and Contributions: This paper outlines the challenge of parallelizing belief propagation, a message passing algorithm for approximate Bayesian inference. It presents a new algorithm based on relaxed priority queues for parallel belief propagation that has probabilistic guarantees of correctness. The algorithm (and variants) are evaluated against a host of baselines and tasks, showing improvements in both speed and efficiency (number of messages).

Strengths: This paper presents a nice combination of theoretical and empirical results and has clear writing, presenting a compelling story backed up by these results. It contextualizes other related works well and provides adequate empirical evidence that they improve upon them. The paper is very accessible and people unfamiliar with belief propagation can learn from the lessons in this paper, namely the novel use of the relaxed priority queue data structure. The theoretical results, though simple, are sufficient to convince readers that the algorithm is correct.

Weaknesses: The paper has one main weakness in my mind. Narratively, the paper does not make a case for belief propagation as an algorithm for modern Bayesian inference. Specifically, it could do a better job contextualizing belief propagation as an alternative to the (current) popular means of approximate inference (e.g. MCMC and variational inference). For the tasks that Relaxed BP is evaluated on in the paper, are MCMC and VI (which are more GPU friendly than BP) viable? This is not clear from the paper. However, as a standalone exploration of BP algorithms, this paper is an interesting read. ---------------------------------- Thank you to the authors for the response which discusses the connections to MCMC/VI. I am also happy with the additional discussion about accuracy and the additional comparisons to other baselines.

Correctness: I didn’t find anything wrong with the proofs/methodology.

Clarity: The paper’s writing is very clear.

Relation to Prior Work: The paper does a good job of comparing to other works and including baselines that reflect the best alternatives.

Reproducibility: Yes

Additional Feedback:


Review 3

Summary and Contributions: In order to have a tradeoff between scalability convergence in developing belief propagation (BP) algorithms, the authors propose to use a relaxed scheduler for priority-based asynchronous BP algorithms s.t. they can be parallelized. They also present positive and worst-case analysis when the problem class is restricted to tree-structured graphical models. Experimental results show that the tradeoff is achieved in comparison with other benchmarks.

Strengths: The problem of exploring the tradeoff between efficiency and convergence is important. The analysis of the proposed relaxed scheduling algorithm on tree structure graphical models is interesting for providing intuitive cases on which the algorithm performs well and not. Experimental results show that the proposed scheme is able to deliver fast convergence and meanwhile keep the number of message updates low.

Weaknesses: - Presentation: I think the space that the paper spends on the BP background is more than necessary since the BP algorithm is just the standard one. The paper would be more compelling if the BP background is compressed and a more complete explanation of their algorithm is presented, for example some visual illustration that comes with the explanation of their implementation in Section 3.3. Moreover, since there are not many notations used in the paper, it is better not to use the same notation for different meanings to avoid confusion. For example, $k$ is used for the number of top elements throughout the paper and also index of variable at Line 285; at Line 301 the parameter $H$ is used without definition, and later on at Line 302 it denotes the tree height while at Line 334 a parameter in the Splash algorithm. - Missing references: There are two highly related work that are not included as baselines. [1] proposed a parallel BP algorithm incorporating belief residual scheduling and uniform work Splash operations; [2] proposed a asynchronous distributed framework to perform BP using a prioritized block scheduler. Could the authors provide some conceptual or empirical comparison of them with the proposed one? - Experiments: One part that is missing in the experiments is the comparison of inference accuracy, in order to show how the relaxed scheduler can affect the accuracy of computed beliefs in comparison with other baselines when they converge. For example in similar way to the Section 6.5 in [2]. [1] Joseph Gonzalez, Yucheng Low, Carlos Guestrin, and David R. O’Hallaron. 2009. Distributed Parallel Inference on Large Factor Graphs. In UAI 2009. [2] Jiangtao Yin and Lixin Gao. 2014. Scalable Distributed Belief Propagation withPrioritized Block Updates. InProceedings of the 23rd ACM International Conferenceon Conference on Information and Knowledge Management, CIKM 2014.

Correctness: The technical details seem correct to me.

Clarity: There is some room for improving the readability of the paper. Detailed suggestions are presented in the weakness part above.

Relation to Prior Work: Some missing references are provided above in the Weakness part.

Reproducibility: Yes

Additional Feedback:


Review 4

Summary and Contributions: The authors investigate the use of 'relaxed' schedulers to effectively parallelize the classical belief propagation (BP) algorithm for marginal inference in discrete graphical models. While BP is an incredibly simple algorithm to understand, its performance is not well understood when applied to arbitrary (loopy) graphical model structures - meaning convergence is not guaranteed and approximation quality can vary wildly. Many methods to improve convergence properties have been proposed and prior work on parallelizing BP has also been proposed. This paper is an incremental update to both of these lines of work - building upon the residual BP algorithm of Elidan and Koller and the parallel splash work of Gonalez et al. The paper is primarily an empirical investigation - the theoretical results appear to heavily leverage the work of Gonzalez et al.

Strengths: The paper is very well written and the few theoretical results appear to be correct, but the paper is primarily an empirical investigation of applying relaxed priority based scheduling to BP updating. The majority of results appear in the appendices, but the few results included in the main paper indicate that the proposed approach works well in some cases. This was an interesting paper to read, but its significance and relevance to the broader NeurIPS community seems limited.

Weaknesses: Given that this was primarily an empirical investigation, I would like to have seen more of the empirical findings be presented up front. The two tables included show some of the promise of the method, but seem a bit incomplete. Why show the speedup without the effect (if any) on accuracy of marginal computation? Also, why do you not vary model instance - e.g. Ising model with different interactions? It seems difficult to draw conclusions based on a single model instance. Can you provide more color on how convergence was assessed for each algorithm? Was the time to assess convergence subtracted from the execution time?

Correctness: As stated above, the few theoretical results seem correct. In the proposed implementation, you state that after a task is chosen it is marked as "in-process" so it cannot be processed by another thread? What happens if the the task chosen by a thread is marked as in-process? How often does this kind of miss occur?

Clarity: Yes. Very well written.

Relation to Prior Work: Yes. The authors thoroughly cover existing art on this topic.

Reproducibility: Yes

Additional Feedback:

[Author Response · NeurIPS 2020]

We thank the reviewers for their comments, and take the opportunity to answer their questions below.

**Positioning relative to MCMC and VI (R2).** This is a very interesting question, and in fact one of our long-term goals
is to perform a rigorous comparison of different inference solutions at scale, using modern parallelization techniques.
Our focus and current experimental data pertain to BP on CPU-centric platforms, and so we will refrain from definitive
statements regarding other methods on specialized hardware.

However, what we can say with good confidence is that BP is still the go-to solution for specific applications such
as LDPC decoding, and that significant work has been invested in speeding up such applications, see e.g. [8] and
references within. Our work should definitely be relevant to such applications, and generally to CPU-centric inference;
conversely, we believe dynamic scheduling ideas can be extended to other modern inference methods on CPUs.

**Writing and notation (R3, R4).** We acknowledge this point, and we will give more prominence to the experimental
evaluation in the next revision. @R3: Thank you for the pointers on notation, which we will address.

**Additional related work (R3).** Thank you for this additional related work, which we will cite and discuss in detail.
The first reference you noted focuses on distribution costs in the multi-node setting (e.g. partitioning the input graphs to
avoid cross-node communication), and is therefore less relevant in terms of direct comparison. However, the second
reference is relevant: we therefore implemented the algorithm in our framework, and present the results below. We take
the $0.1|V|$ best vertices according to the Splash metric, and update the messages from these vertices, as suggested in
this reference. The results in terms of speedup and number of updates relative to the baseline residual BP on one thread
are presented below, where non-baseline algorithms are executed on 70 threads ("—" denotes failure to converge):

Speedup relative to residual BP on 1 thread:

| Model | Relaxed Residual | Synchronous | Yin & Gao |
|---|---|---|---|
| Tree | 1.391x | 2.538x | 1.692x |
| Ising | 6.720x | 3.009x | 3.311x |
| Potts | 7.454x | — | — |
| LDPC | 13.393x | 17.735x | 3.044x |

Number of updates relative to residual BP on 1 thread:

| Model | Relaxed Residual | Synchronous | Yin & Gao |
|---|---|---|---|
| Tree | 1.007x | 48.000x | 5.110x |
| Ising | 1.058x | 45.006x | 3.996x |
| Potts | 1.063x | — | — |
| LDPC | 1.020x | 4.404x | 1.668x |

As can be seen from an examination of the results, the algorithm of [2] has lower performance relative to relaxed
residual on most inputs, due to the consistently higher number of updates.

**Accuracy of inference (R3, R4).** This is a good point, and we will discuss accuracy of inference more explicitly. In
short, we use the LDPC model to test the accuracy of inference. In our LDPC instances, the correct marginals match the
original codeword, and synchronous BP provably recovers these with high probability. All algorithms that converged
recovered the correct codeword, indicating good accuracy for all schedules.

As suggested by R3, we also compared the marginal estimates of the algorithms as in ref. [2] of R3. We computed the
average and maximum differences of point-wise marginal estimates between the baseline residual BP and all other
algorithms. The average difference is less than $10^{-5}$ for all algorithms on LDPC and Ising and within $2 \cdot 10^{-3}$ on Potts.
The maximum difference is less than $4 \cdot 10^{-4}$ for LDPC and Ising, i.e., all algorithms appear to converge to the same
marginals. However, the maximum difference for Potts can be very high, but the residual algorithms seem to converge
to the same answer; this is likely due to the fact that it seems Potts does not always converge properly.

**Model selection and model variety (R4).** We chose the models and parameters following prior work, to provide
reasonable benchmarks and comparison to prior work. The experiments include two model size regimes, with the more
detailed scaling studies in the appendix using a smaller model size. Across our experiments, we found that behavior on
models generated with different random seeds is very similar. Indeed, as our test models are fairly large and drawn
from random ensembles, it is expected that they follow the average behavior of the ensemble as a whole.

**Implementation details on convergence detection and locking (R4).** The convergence thresholds are detailed in
Appendix D. The convergence condition is checked only periodically, as detailed in E.2.2, and has minimal (amortized)
cost, included in the running times.

Tasks that get marked as in-process are withdrawn from the multiqueue, so not other thread can take them. Each thread
takes locks on all messages affected by the task before processing it, in a fixed predetermined order, in order to avoid
deadlocks. Thus, if two threads want to update the same message concurrently, the second one has to wait for the
first to finish. We did not specifically measure the amount of time threads need to wait in this manner; however, we
tested *unfair* versions of the algorithms, where threads can update messages without taking locks (leading to possibly
inconsistent updates). The timing results were virtually identical, suggesting that overlaps are very unlikely in practice.

[Meta-Review · NeurIPS 2020]

Reviewers agreed, in reviews and discussion, that this paper presents a nice, simple idea very clearly. The author feedback included new experiments and a new baseline, with positive results. I enjoyed reading the paper too.